# Value-Directed Remembering: A Dual-Process Perspective

**DOI:** 10.3390/bs15081113

**Published:** 2025-08-17

**Authors:** Qiong Li, Weihai Tang, Xiping Liu

**Affiliations:** 1Faculty of Psychology, Tianjin Normal University, Tianjin 300387, China; jeanieyuqiong@126.com; 2School of Landscape Architecture and Architecture, Jiangxi Environmental Engineering Vocational College, Ganzhou 341000, China; 3School of Sociology, University of Sanya, Sanya 572022, China; twhpsy@126.com

**Keywords:** value-directed remembering, dual-process mechanism, metamemory, memory development

## Abstract

Value-directed remembering involves two key mechanisms: automatic processing and strategic processing. Automatic processing relies on the brain’s reward system and is associated with midbrain dopaminergic pathways and medial temporal-lobe activity. Strategic processing, in contrast, involves conscious, effortful encoding strategies and engages semantic-processing regions and executive control systems. This article reviews the developmental trajectory of value-directed remembering from childhood to old age through the lens of a dual-process model. Children and adolescents primarily rely on automatic processing; adults are capable of flexibly switching between the two processes; older adults tend to rely more on strategic processing. These findings reflect the dynamic developmental changes in the brain’s reward and executive-control systems. Future research should further investigate the synergistic interplay between dual-processing mechanisms, the moderating role of cultural contexts, and the efficacy of intervention strategies to deepen our understanding of the developmental trajectory of value-directed memory.

## 1. Introduction

Human memory is inherently limited in its capacity to encode and retain information. This limitation becomes especially salient in the current era of information overload, where individuals are bombarded with a vast array of content. To achieve efficient memory performance, it is essential to selectively encode and retain information that is most relevant or valuable while ignoring less important content ([8]). This process—prioritizing and deliberately encoding high-value information for future retrieval—is referred to as value-directed remembering (VDR) ([8]; [108]). A growing body of research has demonstrated that individuals can flexibly allocate cognitive resources based on the perceived importance of information, preferentially remembering high-value content ([8]; [90]; [24], [26]).

Value modulates memory performance through two distinct mechanisms: strategic and automatic processing. Strategic processing involves consciously selecting and elaborately encoding important information through deep semantic processing ([14]). This top-down mechanism critically depends on metacognitive skills—particularly the capacity to monitor, select, and adaptively apply optimal encoding strategies ([59]). In contrast, automatic processing operates through bottom-up mechanisms, where memory enhancement occurs when information is intrinsically linked to reward salience or prediction violations. Such effects are generally driven by dopaminergic signaling and prediction error mechanisms in the brain ([47]).

From the perspective of developmental cognitive neuroscience, the interactive mechanisms supporting value-directed remembering demonstrate dynamic reorganization across the lifespan. During childhood and adolescence, memory for valuable information primarily depends on automatic processes—a pattern likely linked to the early development of the brain’s reward system and its preferential modulation of memory encoding. As individuals mature into adulthood, cognitive flexibility increases, allowing for dynamic switching between automatic and strategic modes of processing depending on task demands. In older adulthood, however, strategic processing becomes increasingly dominant, possibly reflecting both a decline in reward-related function and a compensatory reliance on executive control mechanisms ([47]; [111]).

From a dual-process perspective, this paper aims to systematically examine the roles of automatic and strategic processing across different age groups and their underlying neural mechanisms, while also attempting to construct an integrative theoretical framework encompassing mechanisms, development, and methodology (see Figure 1). This framework visually depicts the dual-processing pathways in value-directed memory, their dynamic interactions, lifespan developmental trajectories, and the primary experimental paradigms, providing a theoretical reference for understanding the selectivity and adaptive nature of memory.

## 2. The Dual-Process Mechanism of Value-Directed Remembering

Traditional memory theories have focused on the processes of encoding, storage, and retrieval but have not fully explained why individuals tend to remember some information better than others ([8]). With the advancement of cognitive psychology and neuroscience, researchers have increasingly recognized that memory selectivity is influenced not only by factors such as salience and repetition but also by the subjective value of the information ([47]). Prior research has revealed that two distinct mechanisms—automatic processing and strategic processing—are responsible for the differential encoding of high-value versus low-value information ([47]; [14]; [4]). The brain’s reward system primarily drives automatic processing. When information is associated with potential rewards, this system is automatically activated, enhancing the encoding and retrieval of valuable content ([1]; [11]).

In contrast, strategic processing involves deliberate, deep semantic encoding strategies and engages regions associated with semantic elaboration and executive control ([15]). Based on these findings, researchers have proposed a dual-process theoretical framework to explain both the neural and behavioral aspects of value-directed remembering.

### 2.1. Automatic Processing

Automatic processing refers to the spontaneous encoding, storage, and retrieval of information with relatively low cognitive-resource consumption. In the context of value-directed memory, this process is primarily regulated by neural mechanisms driven by reward prediction error (RPE) ([47]). Midbrain dopaminergic neurons dynamically modulate synaptic plasticity by encoding RPE (RPE = actual reward − expected reward): a positive RPE, accompanied by phasic dopamine release, significantly enhances encoding efficiency, whereas a negative RPE suppresses memory processing ([86]). In human declarative memory studies ([104]), positive RPE selectively enhances activity within the ventral tegmental area (VTA)–hippocampal circuit, facilitating long-term consolidation of unexpectedly high-reward items ([33]). Under this mechanism, high-value information preferentially activates the striatal nucleus accumbens network, which not only encodes reward salience ([48]) but also participates in value computation based on reinforcement learning principles ([22]), thereby biasing attention and memory toward high-value items. When individuals anticipate high rewards, VTA-hippocampal functional connectivity is enhanced, directly optimizing memory storage ([1]).

The spontaneous nature of automatic processing is particularly evident in involuntary autobiographical memories (IAMs). According to the direct retrieval theory, IAMs are triggered by bottom-up, automatic associative activation rather than conscious control ([3]). Developmental research indicates that IAMs emerge early in life and maintain a relatively stable frequency across age, contrasting with the gradual increase in voluntary autobiographical memory with age. For instance, young children often spontaneously recall past experiences in response to specific environmental cues, such as particular smells or scenes, demonstrating that automatic memory-retrieval mechanisms are already functional before the maturation of executive control.

Automatic processing in value-directed remembering not only relies on the coordinated activity of the reward system and memory-related brain regions but is also closely linked to motivation. As an intrinsic driver of behavior, motivation determines the allocation of attention, the prioritization of information processing, and memory storage ([69]; [105]). High motivational states can significantly enhance preferential processing of high-value information, even under unconscious conditions. [78] ([78]) found that participants’ behavioral responses were amplified for high-value stimuli even when they were not consciously aware of them, indicating that reward effects can occur without conscious awareness. Similarly, [6] ([6]) demonstrated that individuals performed better on high-reward tasks than on low-reward tasks under unconscious conditions. These findings provide compelling evidence for the central role of motivation in automatic processing.

Rewards not only enhance memory for directly associated information but can also strengthen the encoding of stimuli occurring before or after the reward via a retroactive effect ([5]). Motivation is generally classified into extrinsic (e.g., monetary or material rewards) and intrinsic (e.g., autonomy, sense of achievement) types ([23]). Extrinsic rewards can facilitate memory during both encoding and consolidation phases ([110]).

During the encoding phase, reward anticipation preferentially triggers automatic processing, primarily dependent on activation of the brain’s reward system ([1]). Using fMRI combined with computational modeling, [17] ([17]) revealed how multiple brain regions dynamically participate in value computation and decision guidance. Activity patterns in the caudate, amygdala, and orbitofrontal cortex adjusted dynamically according to reward expectation and reward prediction error (RPE). Furthermore, individual differences in reinforcement learning parameters, such as learning rate and risk preference, significantly enhanced the prediction of both behavior and neural activity. Subsequent studies further elucidated mechanisms of value integration. [31] ([31]) identified the ventromedial prefrontal cortex (vmPFC) as a key hub in value-based decision-making, responsible for integrating behaviorally derived value signals. A meta-analysis by [29] ([29]) highlighted the striatum as a core region for encoding prediction errors, with distinct neural representations for reward versus punishment: reward prediction errors were concentrated in the striatum, whereas punishment prediction errors involved the insula and pallidum.

During the consolidation phase, reward-induced dopamine signals can directly act on memory-related structures, such as the hippocampus, to enhance the stability of long-term memories ([77]). By contrast, research on the mnemonic effects of intrinsic motivation is comparatively limited. [70] ([70]) demonstrated that even in the absence of explicit rewards, active decision-making significantly reduced forgetting rates in both immediate and delayed (24 h) tests, accompanied by increased striatal activation and enhanced post-encoding hippocampus–perirhinal cortex (PRC) functional connectivity. This suggests that autonomous choice can support memory consolidation through the striatum–hippocampus–PRC network, thereby improving long-term memory performance.

However, value-directed remembering does not rely solely on automatic processing. Value evaluation often requires integrating multidimensional information, including emotional and social cues, which necessitates the deep involvement of strategic processing. Moreover, goal-directed behavior depends on strategic processing for monitoring and adjustment, thereby enhancing adaptability and flexibility ([88]; [67]). Consequently, in value-directed memory, automatic and strategic processes operate in a complementary and synergistic manner.

### 2.2. Strategic Processing

In value-directed remembering (VDR), strategic processing represents a critical cognitive mechanism by which individuals intentionally employ specific encoding strategies to prioritize and remember high-value information, thereby enhancing both memory efficiency and performance. When presented with high-value items, individuals are more likely to engage in deep semantic encoding, forming meaningful contextual associations that increase memory strength and familiarity while also facilitating semantic understanding. Compared to automatic processing, strategic processing has a more substantial effect on the memory of high-value information ([14]). This deliberate engagement of semantic elaboration not only optimizes memory performance but also plays a pivotal role in supporting cognitive decision-making when confronted with complex information.

A growing body of research has demonstrated that individuals actively apply mnemonic strategies to prioritize high-value items, leading to significantly better recall compared to low-value items ([96]; [73], [72]; [42]; [102]; [62], [63]; [67]; [91]). This selective encoding advantage is closely linked to metacognitive abilities ([89]; [60]; [59], [65]; [66]; [58]; [91]). Individuals with stronger metacognitive insight are better able to assess their memory capacity and adaptively modify their encoding strategies to optimize performance. As task experience and feedback accumulate, the ability to selectively remember high-value information improves significantly ([2]; [87], [88]). For instance, [10] ([10]) demonstrated that under time-constrained learning conditions, participants successfully prioritized high-value items by strategically allocating greater cognitive resources to their encoding. This selective investment of study time resulted in optimized memory performance for the most valuable information.

Neuroimaging studies have shown that strategic processing in VDR is associated with increased activation in several key brain regions, including the ventrolateral prefrontal cortex (VLPFC), pre-supplementary motor area (pre-SMA), and posterior lateral temporal cortex ([16]). These regions are critically involved in deep semantic elaboration. Among them, the VLPFC plays a central role as part of the executive control network. Recent findings further suggest that successful strategic encoding of high-value information depends on the coordinated activation of semantic-processing regions and executive-control systems. [39] ([39]) demonstrated that enhanced semantic processing significantly improves memory for high-value items. When individuals are constrained to remembering only a subset of items, they tend to selectively allocate attention to the most valuable ones while ignoring less valuable content, thereby maximizing overall memory output. Strategic processing also entails prioritized refreshing and attentional allocation toward high-value information during working-memory maintenance. Recent research has shown that high-value items are more likely to be refreshed in working memory, not by increasing the time spent on each item but by increasing the frequency with which these items are refreshed ([50]).

While strategic and automatic processing relies on different underlying neural systems, they are both essential to value-guided memory. Strategic encoding of high-value content is associated with increased activation in left-hemispheric regions involved in semantic processing, supporting the role of elaborative encoding in this process ([47]). In contrast, automatic processing depends more on the activation of the midbrain dopaminergic reward system, which selectively enhances encoding through dopamine-mediated signals ([33]). However, [47] ([47]) emphasized that these mechanisms are not mutually exclusive; rather, encoding in value-directed remembering reflects a dynamic interaction between strategic and automatic processes.

### 2.3. The Interaction Between Automatic and Strategic Processing

Automatic and strategic processing in value-directed remembering are not independent mechanisms but operate with close interaction. Automatic processing of high-value information may lay the groundwork for subsequent strategic processing, further amplifying memory performance ([15]). Specifically, when the value of an item exceeds an individual’s expectations, these high-value items automatically capture attention, thereby prompting the allocation of greater attentional resources and the selection of optimal memory strategies to ensure their effective encoding and retrieval ([47]).

Research on prospective memory offers a unique perspective for understanding the interactive mechanisms between automatic and strategic processing within the memory system. The classic study by [95] ([95]) on intentional reminding behavior in young children elucidates the developmental trajectory of this synergistic mechanism. Their findings revealed that children aged 2 to 4 demonstrated a significant advantage in high-value tasks (e.g., “remind to buy candy”), achieving up to 80% accuracy in unprompted recall. First, even before the full maturation of the prefrontal cortex, high-value intentions received prioritized encoding via reward systems such as the striatum, reflecting the foundational role of automatic processing within memory. Second, the study observed that 4-year-olds outperformed 2-year-olds in low-interest tasks, indicating that with the progressive maturation of prefrontal cortex functions, the capacity for intention maintenance was markedly enhanced—an early sign of strategic processing development. The multiprocess theory of prospective memory proposed by [52] ([52]) further supports a dual-processing account: on the one hand, when target cues are salient or highly relevant to the current context, automatic processing dependent on the striatum–hippocampal circuitry enables rapid intention retrieval; on the other hand, for non-salient cues, strategic monitoring mediated by the frontoparietal network requires sustained cognitive resources and impacts concurrent task performance. These findings demonstrate that a single mechanism does not support prospective memory but instead constitutes a dynamic system underpinned by automatic triggering and strategic monitoring.

In the field of value-directed memory, both behavioral and neuroimaging evidence support the notion of dual-process cooperation. [14] ([14]) manipulated retrieval conditions. They found that experience with free-recall testing during the encoding phase not only enhanced subsequent recognition of high-value information but also facilitated detail recollection dependent on prefrontal control and familiarity-based retrieval reliant on the dopaminergic system. The underlying mechanisms may include activation of the prefrontal–hippocampal circuit during free recall, prompting participants to adopt more refined semantic encoding, while the VTA–striatum–hippocampus dopaminergic pathway strengthens the representational stability of high-value items. Metacognitive monitoring plays a regulatory role in this process: when participants recognize limitations in their memory through test feedback, they proactively adjust attention allocation, prioritize high-value information, and further engage the reward system’s automatic response.

[39] ([39]) provided additional evidence for dual-process cooperation. Participants studied value-tagged words and subsequently received “remember” or “forget” cues. Results indicated that for high-value items, “remember” cues significantly enhanced memory performance; however, even under “forget” cues, recognition sensitivity for high-value information remained higher than for low-value items, suggesting that high-value cues can trigger automatic processing even in the absence of external reinforcement.

Neuroimaging studies offer direct support for this cooperative mechanism. [15] ([15]) observed that encoding high-value information was associated with increased activation in the left inferior frontal gyrus and left posterior lateral temporal cortex—regions linked to deep semantic processing—indicating a central role of strategic processing in high-value encoding. Simultaneously, modest activation in the midbrain and ventral striatum correlated with memory performance, demonstrating the contribution of automatic processing within strategic encoding. [83] ([83]), using an fMRI monetary incentive delay (MID) task, revealed a hierarchical neural mechanism for reward processing: the subcortical reward regions (ventral striatum and ventral tegmental area) responded to reward cues independently of attentional tasks, displaying typical automatic features. The higher-order visual cortex (fusiform gyrus) integrated input from both subcortical reward pathways and frontoparietal attention networks, reflecting dual-pathway modulation; in contrast, reward processing in the anterior insula and anterior cingulate cortex was attention-dependent, showing enhanced activation only when attention was directed toward reward cues. Dynamic causal modeling indicated that such modulation is implemented via attentional regulation of ventral striatum–anterior insula connectivity. [30] ([30]) further demonstrated that reward anticipation significantly enhances prefrontal cortex activity, optimizing core executive functions such as response inhibition and cognitive control, thereby providing critical neural support for the cooperative operation of strategic and automatic processing.

These theoretical reviews, behavioral experiments, and neuroimaging findings illuminate the cooperative mechanisms between automatic and strategic processing in value-directed remembering. They provide robust theoretical and empirical support for understanding their complementary and interactive roles in memory formation.

## 3. Development of Value-Directed Remembering

The capacities for automatic and strategic processing exhibit marked differences across developmental stages. From childhood to adolescence, automatic processing abilities show substantial enhancement, closely linked to the maturation of the midbrain–dopaminergic system. This developmental change is accompanied by increased sensitivity to reward signals, thereby promoting the prioritized encoding and storage of high-value information ([28]). Neuroimaging meta-analyses indicate that reward-related regions, such as the ventral striatum, reach peak activation during adolescence ([92]). Encoding studies in children further demonstrate that reward contexts can enhance memory sensitivity ([71]) and that reward processing and associative memory capacities undergo dynamic changes throughout development ([54]; [13]). Behavioral evidence suggests that reward-seeking behaviors follow an inverted U-shaped trajectory during adolescence, with dual peaks observed around ages 12–15 and 17–18 ([97]; [93]). Enhanced reward sensitivity during adolescence not only affects cognitive performance but may also relate to changes in depression risk ([82]). For example, neural sensitivity to eudaimonic rewards may buffer depressive symptoms ([98]). Moreover, the interactive dynamics among reward-system function, life-stress exposure, and emotional states during adolescence may differ from those in early adulthood ([18]). Recent evidence indicates that ventral-striatum and amygdala activation during reward anticipation can modulate the association between life-stress and depressive symptoms in adolescents, suggesting that heightened neural-reward sensitivity may buffer the adverse emotional effects of stress ([27]).

In contrast, the development of strategic processing is relatively protracted. During childhood and early adolescence, the prefrontal cortex is not fully mature, and higher-order cognitive functions such as inhibitory control and executive regulation are limited, making reward processing more likely to interfere with strategic information processing ([7]; [75]). Structural immaturity may also reduce the efficiency of value-evaluation systems ([21]). In late adolescence, structural and functional maturation of the prefrontal cortex supports significant improvements in executive control, thereby facilitating the development of strategic processing ([74]). Nevertheless, compared to adults, adolescents still exhibit deficits in the selection and implementation efficiency of memory strategies ([94]). Cross-sectional studies show that children (5–9 years), adolescents (10–17 years), and young adults (18–23 years) all demonstrate a priority recall effect for high-value information; however, children and adolescents display significantly lower selection accuracy than young adults ([9]). Longitudinal evidence indicates that Selective Learning Efficiency (SLE) steadily increases with age, likely reflecting brain structural maturation ([35]; [79]). From a neural perspective, the functional integration of frontal–basal ganglia circuits is considered a core mechanism underlying the development of inhibitory control, providing a critical neural foundation for the maturation of strategic processing ([43]).

Age-related dynamic studies further elucidate the neural mechanisms underlying memory development in children and adolescents. [13] ([13]) examined participants aged 8–25 years. They found that enhanced functional connectivity between the dorsolateral prefrontal cortex (dlPFC) and the ventral tegmental area (VTA) during the encoding phase was significantly associated with improvements in high-reward item-specific memory, with this association increasing with age. Changes in connectivity between the anterior hippocampus and VTA were more strongly linked to gains in high-reward gist memory among children and adolescents. Adolescents exhibited superior performance in high-reward general-source memory compared to both children and adults, but their performance under low-reward conditions was comparatively weaker. These findings suggest that heightened reward sensitivity during adolescence may selectively facilitate or interfere with memory performance depending on the context, and they also highlight the potentially greater role of post-encoding interactions within the mesolimbic system in the formation of reward-based memories in children ([49]).

Adulthood marks the peak of both automatic and strategic processing abilities. Functional connectivity between the midbrain dopaminergic system and medial temporal regions becomes more stable, significantly enhancing memory for high-value information. Functional MRI studies indicate that the reward system can rapidly evaluate the value of information through automatic processing mechanisms and modulate memory systems to optimize encoding of high-value items ([1]). Functional connectivity between the ventral striatum and hippocampus is strengthened during adulthood, supporting prioritized encoding and retrieval of high-value information ([7]; [20]). In addition, adults demonstrate high cognitive-control flexibility, allowing them to dynamically adjust memory strategies according to task demands—for example, engaging in deep semantic processing when encoding important information ([39]; [47]). Experimental evidence further shows that when participants are instructed to use a uniform strategy for all items (e.g., mental rehearsal or imagery), memory for low-value items is also enhanced ([39]). Thus, the encoding advantage for high-value information arises not only from motivational drives but also from strategic-processing involvement.

In older adulthood, prefrontal cortical volume gradually declines ([80]), accompanied by substantial reductions in memory and cognitive function ([84]; [99]). Functional decline in the midbrain dopaminergic system leads to decreased automatic-processing capacity, impairing the enhancement of memory for important information ([85]). Nevertheless, despite deterioration in neural-reward mechanisms and prefrontal function, older adults continue to exhibit selective memory for high-value information. This phenomenon can be explained by the Socioemotional Selectivity Theory (SST) ([81]), which posits that as perceived future time diminishes, older adults adjust their goal systems to prioritize emotional regulation over knowledge acquisition. Consequently, they tend to preferentially select and maintain positive information, producing the characteristic “positivity effect” ([44]). Neurobehavioral evidence indicates that this effect is both delayed and context-dependent, manifesting as sustained attention to positive stimuli and inhibition of negative stimuli, particularly under free-recall conditions ([41]; [46]; [45]).

At the level of cognitive processing, although automatic processing declines, older adults display adaptive strategic processing and metacognitive regulation ([36]; [61]). They leverage accumulated semantic knowledge and life experience, employing selective attention and flexible memory strategies to prioritize encoding of high-value information relevant to current goals. This optimization of cognitive resources, closely linked with metacognitive awareness, enables older adults to manage cognitive load and maintain task-performance effectively ([89]).

## 4. Research Methods in Value-Directed Memory

In the study of value-directed remembering, free recall and recognition tests represent two core-memory measurement paradigms, revealing significant mechanistic differences in how the memory system processes value-related information. Much empirical research shows that free recall primarily relies on controlled, strategic processing, enabling individuals to prioritize the encoding and retrieval of high-value information selectively. In contrast, recognition tasks depend more heavily on automatic processing, with neural mechanisms predominantly involving dopaminergic reward-system modulation ([14]). This mechanistic distinction results in different manifestations of value effects across paradigms: the facilitative impact of value on memory is pronounced in free recall, whereas it tends to be comparatively weaker in recognition tasks ([57]).

### 4.1. Free-Recall Tests

The most commonly employed paradigm within free-recall research is Value-Directed Remembering (VDR). A typical VDR task consists of three stages. First is the learning phase, during which participants are presented with study materials paired with point values (e.g., “television—8”) and are informed that they will earn corresponding points for each correctly recalled item, to maximize their total score. Second, the interference phase introduces a filler task before recall to eliminate short-term memory effects. Third, the recall phase requires participants to retrieve as many studied items as possible, with total scores computed accordingly. Measurement typically involves the number of items recalled and the Selectivity Index (SI), which quantifies an individual’s sensitivity to high-versus low-value information. The Selectivity Index (SI) is mathematically expressed as: SI = (Actual Score − Chance Score)/(Ideal Score − Chance Score).

For example, if the study list contains 12 words with values ranging from 1 to 12 points, and a participant recalls six words valued at 12, 10, 9, 7, 6, and 4 points, respectively, the actual score is 48 (12 + 10 + 9 + 7 + 6 + 4). The ideal score is the sum of the highest six values, 57 (12 + 11 + 10 + 9 + 8 + 7). The chance score equals the average value (6.5) multiplied by the number of recalled items (6), resulting in 39 (6.5 × 6). Substituting these into the formula yields an SI of 0.5. The SI ranges from −1 to 1, where values closer to 1 indicate stronger selectivity for high-value items, values near 0 indicate no value-directed memory, and values approaching −1 indicate a bias toward recalling low-value items.

Researchers frequently manipulate the value distribution of study materials to examine the effect of value gradients on memory. These manipulations include binary high–low structures (e.g., 1 vs. 10 points) ([65]; [76]), ternary low–medium–high structures (e.g., 1, 5, 10 points) ([53]), repeated 1–10 point structures (each value presented twice) ([55]; [88]), fully continuous 1–20 point structures (each value paired with a unique item) ([68]; [66]), and other value arrangements.

For data analysis, analysis of variance (ANOVA) is commonly employed to compare overall selectivity. In contrast, multilevel modeling (MLM), treating value as a continuous variable at the trial level, captures individual differences more effectively and allows precise assessment of value effects in complex data structures ([57]).

### 4.2. Recognition Tasks

Recognition tests in value-directed memory research commonly employ the Monetary Incentive Encoding (MIE) paradigm or recognition tasks adapted from the Value-Directed Remembering (VDR) framework. These experiments typically include three phases: reward-cued encoding, an interference task, and a recognition test, during which participants make “old/new” judgments. Recognition tasks are often combined with Remember/Know (R/K) judgments to dissociate recollection-based from familiarity-based processing ([100]). Evidence indicates that high-value information primarily enhances recollection, with relatively limited effects on familiarity ([25]; [38]).

To improve measurement precision, the Recognition Without Cued Recall (RWCR) paradigm, combined with signal detection theory, allows for more accurate quantification of recollection and familiarity components while reducing error accumulation ([12]). Additionally, Receiver Operating Characteristic (ROC) curve analysis is frequently used to examine the dual-process characteristics of memory; asymmetries in the ROC curve reflect the joint contributions of recollection and familiarity.

Metrics for recollection (R) and familiarity (F) are typically derived from the independent process model proposed by [109] ([109]), with corrections for guessing by subtracting false alarm rates. Two approaches are commonly applied for handling false alarms. The first is the overall false alarm method, which assumes that false alarm rates for new high- and low-value items are equal; this approach is simple but may underestimate false alarms for high-value items. The second is the value-specific false alarm method, which infers false-alarm rates for each category based on the distribution of hits. This method aligns better with the assumption that high-value items are encoded more thoroughly, but it requires that the likelihood of misidentifying new versus old items is symmetric across value categories; otherwise, recollection and familiarity estimates may be biased. Researchers often conduct sensitivity analyses to assess the robustness of results across different false-alarm correction methods, thereby ensuring the reliability of memory-parameter estimates.

In summary, free recall and recognition tasks each offer distinct measurement advantages. By integrating multilevel analyses with methodological refinements, these paradigms can systematically elucidate the contributions of strategic and automatic processing in value-directed memory.

## 5. General Discussion and Future Directions

This review examined the developmental trajectory of value-directed remembering (VDR) through the lens of dual-process mechanisms, highlighting age-related differences in the roles of automatic and strategic processing. Prior research has consistently shown that both processing modes contribute to enhanced memory for high-value information ([63], [64]). However, due to differences in the maturation timelines of relevant brain regions, children and adolescents tend to be less effective at strategically encoding high-value items compared to adults. Nonetheless, their ability to selectively remember valuable information improves steadily throughout adolescence, reaching full maturity in adulthood ([9]; [13]). As individuals age further, memory encoding and retrieval capacities decline. Yet, older adults maintain the ability to selectively remember important information by relying on preserved cognitive control, metacognitive skills, and life experience ([9]; [89]). While substantial progress has been made, several critical questions remain. Future research can address these gaps through the following directions:

### 5.1. The Interaction of Dual-Process Mechanisms in Adolescents and Adults

Although previous studies have often examined automatic and strategic processes in isolation, recent empirical findings suggest that the memory advantage for high-value items results from their dynamic interaction ([47]). For example, in adults, strategic processing may exert a stronger influence on memory than automatic mechanisms ([14]). By contrast, adolescents often outperform adults on reinforcement learning tasks, possibly due to heightened connectivity between reward evaluation and memory systems during puberty ([20]). Future research should examine how these mechanisms interact across age groups to influence value-based memory selectivity and neural-activation patterns. Multimodal neuroimaging techniques such as fMRI and EEG could elucidate the underlying neural dynamics at different developmental stages.

### 5.2. Neural and Cognitive Mechanisms of VDR in Older Adults

Selective encoding of valuable information is closely linked to metacognitive functioning. When presented with items of varying value, individuals tend to allocate more cognitive resources to higher-value content ([9]). Despite age-related declines in general memory capacity, healthy older adults can use metacognitive strategies to maintain selective memory performance ([89]). However, patients with neurodegenerative conditions, such as behavioral variant frontotemporal dementia (bvFTD), often exhibit impairments in value-based encoding. [107] ([107]) suggested that reduced motivational responses in bvFTD may be partially due to reliance on point-based reward systems. Future research should explore whether more tangible incentives (e.g., monetary rewards) and high-resolution neuroimaging can reveal how different reward types influence VDR in populations with cognitive decline.

### 5.3. The Shaping Role of Cultural and Contextual Factors on Dual-Processing Mechanisms

Cultural values shape memory-processing patterns from early developmental stages ([103]). Western individualistic cultures reinforce an independent self-construal, promoting the development of goal-directed, detail-oriented strategic memory processing, which at the neural level manifests as enhanced control in the dorsolateral prefrontal cortex ([37]). In contrast, East Asian collectivistic cultures emphasize the automatic processing of social context and background information, with memory content often reconstructed semantically and relationally; this is closely associated with increased activation of the default mode network ([34]). Such differences may stem from distinct early perceptual processing strategies ([19]; [56]) and exhibit a degree of plasticity in response to changes in task context ([34]). Future research should further explore the interaction between cultural background and developmental stages to elucidate the adaptation and transformation of value-directed memory across cross-cultural contexts.

### 5.4. Developmentally Informed Intervention Strategies for Value-Directed Memory

Age-related differences are evident in memory performance and influence intervention efficacy. Children and adolescents, due to incomplete prefrontal-cortex development and limited metacognitive capacities, show markedly lower selective memory than adults ([9]). Recent findings indicate that test-feedback experience can significantly enhance memory selectivity in value contexts ([51]). Although adults and older adults possess stronger strategic processing abilities, their attentional resource allocation is susceptible to disruption by emotional states and stress ([47]). Therefore, integrative interventions combining value cues with stress management may be more effective. Large-scale projects such as the ACTIVE study ([106]) and meta-analyses ([40]; [32]; [101]) have confirmed that systematic strategy training substantially improves memory performance in older adults. Integrating these training approaches within the Value-Directed Remembering (VDR) framework holds promise for optimizing memory selectivity across different age groups. Nevertheless, the field lacks long-term follow-up, transfer-effect validation, and fine-grained neural mechanism modeling.

### 5.5. Directions for Methodological and Experimental Design Optimization

Future VDR research requires further methodological refinement to enhance ecological and internal validity. For example, assigning value labels to old and new items in recognition paradigms could reduce systematic biases in false-alarm-rate estimation. Analytically, incorporating multilevel linear modeling (MLM) allows simultaneous handling of item-level and individual-level variables, while Bayesian methods provide robust parameter estimation and credible intervals. Combining signal detection theory (SDT) with reinforcement learning models can elucidate the dynamic influence of value on memory encoding and retrieval processes. Furthermore, integrating neurophysiological data such as eye-tracking, fMRI, or EEG can facilitate the construction of computational maps of value-modulated memory pathways, clarifying the neural implementation of reward and motivation across different processing stages and providing empirical support for refining the VDR theoretical framework.

## Figures and Tables

**Figure 1 behavsci-15-01113-f001:**
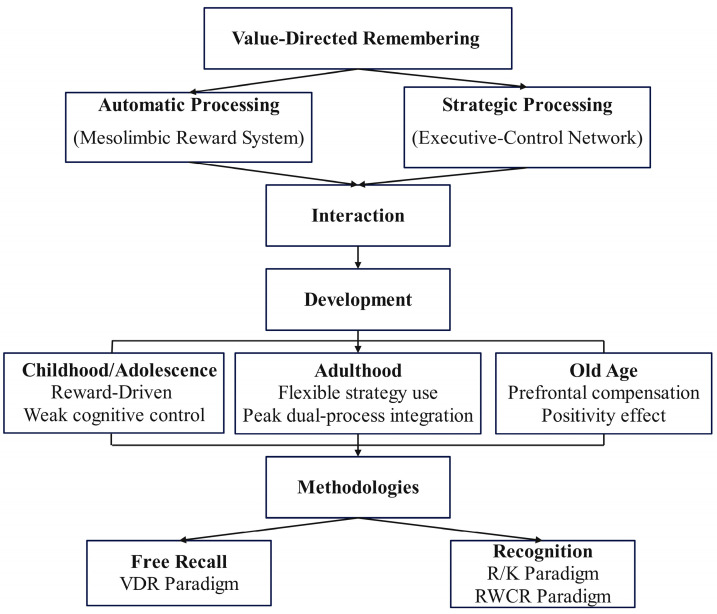
A dual-process framework of value-directed memory. The upper part of the figure illustrates the two core memory-processing pathways: automatic processing, which relies on the midbrain–limbic reward system, and strategic processing, which depends on the executive-control network. These two mechanisms exhibit dynamic interactions across the lifespan: during childhood and adolescence, reward-driven automatic processing predominates due to limited cognitive control; in adulthood, strategic processing shows peak flexibility and optimal integration of the dual processes; in older age, automatic processing declines, leading to greater reliance on frontal executive networks for compensation, accompanied by the positivity effect. The lower part of the figure summarizes the two main experimental paradigms: free recall tasks (value-directed remembering, VDR) primarily assess strategic processing, whereas recognition tasks (e.g., Remember/Know paradigm and Rewarded Word-Classification-Recognition, RWCR paradigm) allow the dissociation of recollection and familiarity, providing experimental tools for investigating the dual-process mechanisms.

## Data Availability

No new data were created or analyzed in this study.

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
