# Peer review of "Value-Directed Remembering: A Dual-Process Perspective"

_behavsci, 2025, doi:10.3390/bs15081113_

Round 1

Reviewer 1 Report

Comments and Suggestions for Authors

Review Report for The Manuscript behavsci-3763785

General Comments

Dear Editors,
I have reviewed the manuscript titled “Value-Directed Remembering: A Dual-Process Perspective”. This review examines how value-directed remembering (VDR) relies on two interacting mechanisms: automatic processing and strategic processing. Beyond discussing the differences between automatic and strategic processes, highlighting the brain's reward system's role in automatic processes and conscious encoding and metacognitive control in strategic processes, the review also addresses evidence from neuroimaging, behavioral studies, and lifespan research to support the dual-process model. Although the topic is appealing for the audience of the journal, at the present stage, the review lacks a coherent, critical synthesis in many of its parts, and there are some concerns and limitations that need to be carefully addressed. In addition, certain areas of the literature have not been sufficiently acknowledged, and some inaccuracies should be revised. Addressing these concerns will improve the manuscript’s contextualization and accuracy, and I trust that the author will take this step to strengthen the quality of their work. These points are more specifically detailed below.

Detailed Comments

Methodological limitations

The paper appropriately highlights some evidence supporting the dual-process framework, addressing both reward-related dopaminergic activation and the role of semantic elaboration and metacognitive monitoring. However, one of the major concerns is the overall focus and structure of the manuscript: there are several sections that deserve careful attention and that would benefit from restructuring with a greater conceptual depth. Many paragraphs repeat similar ideas and concepts without further elaborating specific details, e.g. on how the two processes interact dynamically in different contexts, etc. Another issue is that the paper does not critically evaluate possible inconsistencies in the literature or methodological limitations of the state of the art, which should be critical aspect of a review. Many statements (e.g. about the effectiveness of interventions etc..) would benefit from clearer evidence or examples. Additionally, incorporating an overview diagram that summarizes the review's structure and the models it addresses could enhance the visual presentation, favouring readability.

State of the art and References

The manuscript would certainly improve with a more thorough discussion of existing literature. Given the subject's extensive and interdisciplinary nature, the current number of references (69) appears limited for a review paper. Numerous contributions from theoretical, behavioral, computational, and experimental fields have influenced our comprehension of the reward system and reward representation under various perspectives. Although significant research is included, crucial literature—especially regarding modeling studies and systems-level perspectives on anticipatory reward processing—is absent (
https://doi.org/10.1016/j.neuroimage.2015.07.083 ; https://doi.org/10.1126/science.275.5306.1593 ,https://doi.org/10.1038/nature04766 ) . These encompass results from behavioral investigations, neuroimaging, and invasive electrophysiological studies, which are crucial (https://doi.org/10.1093/scan/nsl021 , https://doi.org/10.1016/j.neubiorev.2013.03.023; https://doi.org/10.1002/hbm.22383; https://doi.org/10.1016/j.dcn.2011.06.004; https://doi.org/10.3389/fpsyg.2023.1125066; https://doi.org/10.3389/fpsyg.2022.651547 https://doi.org/10.1016/j.jad.2023.02.149 ; https://doi.org/10.1093/cercor/bhn098 ; https://doi.org/10.1073/pnas.1323014111).

These should be integrated either in Section 2 or 4 for a more thorough understanding. This would establish a more robust connection between empirical evidence and theoretical frameworks. ensuring an appropriate manuscript's contextualization.

Author Response

Dear Reviewer,

We sincerely thank you for your time and effort in reviewing our manuscript and for providing insightful and constructive comments. Your suggestions have been invaluable in guiding us to improve the manuscript’s structure, deepen the theoretical framework, and optimize the research context. We have carefully addressed all comments and made comprehensive revisions accordingly. Below, we provide detailed responses to each point raised.

Comment 1: Several sections require careful revision and more profound conceptual elaboration. Many paragraphs contain repetitive ideas without further explanation, such as how dual processes interact dynamically under different contexts.

Response: We fully agree with this point. We carefully reviewed the manuscript to remove redundancies and supplement and deepen the relevant content. In Section 2 (pages 6–8, lines 228–328), we added empirical studies elaborating on the dynamic interaction between strategic and automatic processing, enhancing conceptual depth and coherence.

Comment 2: Strengthen methodological critique.

Response: We have added a more detailed introduction and critical analysis of research methodologies in Section 4 (pages 10–14, lines 469–669).

Comment 3: Include more empirical evidence or case examples supporting the intervention section.

Response: Relevant empirical studies have been incorporated into Section 5 (page 16, lines 725–741) to reinforce the intervention discussion.

Comment 4: Add an overview figure to improve the manuscript’s logical clarity.

Response: We have included Figure 1(page 2) in the revised manuscript. It presents an integrated framework that covers theoretical background, dual-process mechanisms, developmental trajectories, and research methods to enhance overall logical flow and visualization.

Comment 5: Supplement references to cover a broader range of fields.

Response: Thank you for the valuable literature recommendations. We expanded the reference list from 69 to 105 citations. Among your suggested references, we have selected and incorporated four key papers, cited on pages 4 (lines 131–142; 142–150), 7 (lines 306–324), and 8 (lines 342–356). These enrich the theoretical, behavioral, and computational modeling perspectives on reward systems and representations. We will continue to study other recommended works for future research development.

We believe these revisions have substantially improved the manuscript regarding theoretical integration, and methodological rigor, thus better aligning it with the journal’s scope. Once again, we sincerely thank you for your thorough and thoughtful evaluation. Your constructive feedback has been crucial for enhancing the quality of our work.

Sincerely,
Qiong Li

Reviewer 2 Report

Comments and Suggestions for Authors

I enjoyed reading this paper, which is a well written and informative review of a relatively small body of research on the dual-process perspective of value-directed remembering. The paper also proposes directions for future research.

As a developmental psychologist, I would be interested to see whether and how the authors think this perspective relates to some well-documented phenomena in memory development, as follows.

First, can they comment on the age-related positivity effect, that is, the phenomenon whereby older adults preferentially remember positive- over negative information, while for younger adults the trend is reversed? According to socio-emotional selectivity theory, the effect reflects a combination of controlled and automatic processes (Reed & Carstensen, 2012).

Second, can they comment on findings that involuntary autobiographical memory (IAM) is evident from very early in life, with the frequency of IAMs appearing to be relatively age-invariant - in contrast to marked age-related gains in voluntary recall of autobiographical memories (Berntsen, 2024)?

Third, are there any links with prospective memory (i.e., remembering to do something in the future)? For example, very young children (2- and 3-year-olds) typically perform poorly on laboratory-based prospective memory tasks. However, in real-world tasks with high motivational appeal they are capable of impressive levels of accuracy (Somerville et al., 1983).

Author Response

Dear Reviewer,

We sincerely thank you for your time and effort in reviewing our manuscript and for providing insightful and constructive comments. Your suggestions have been invaluable in guiding us to improve the manuscript’s structure, deepen the theoretical framework, and optimize the research context. We have carefully addressed all comments and made comprehensive revisions accordingly. Below, we provide detailed responses to each point raised.

Comment 1: Regarding positivity effects related to aging.

Response: We added relevant discussion in the developmental section (page 10, lines 440–457), analyzing older adults’ tendency to prioritize positive information from the perspective of socioemotional selectivity theory, and integrating the dual-process framework to explore the roles of strategic processing and motivational regulation.

Comment 2: On involuntary autobiographical memories (IAMs).

Response: A new discussion on IAMs was added on page 3 (lines 98–109), considering its potential significance in value-directed memory research.

Comment 3: On prospective memory (PM).

Response: We supplemented examples of PM research in the interaction mechanisms section (page 6, lines 236–256), highlighting evidence of high accuracy in young children under high motivation conditions and discussing its relation to automatic and strategic processing interaction.

Once again, we sincerely thank you for your thorough and thoughtful evaluation. Your constructive feedback has been crucial for enhancing the quality of our work.

Sincerely,
Qiong Li

Round 2

Reviewer 1 Report

Comments and Suggestions for Authors

Second Review Report for Manuscript ID: behavsci-3763785

I sincerely thank the authors for the effort they have put into addressing my first-round comments and concerns. I appreciate the work invested in revising the manuscript, which has already improved in clarity and scope. However, having reviewed the revised version, I still believe some additional work is necessary at this stage before the paper is ready for final publication. I provide more details below.

Detailed Comments

  • As I stated in my first report, I did not request any substantial restructuring, new analyses, or foundational methodological revisions. Therefore, my expectation was that my comments would be addressed more thoroughly, particularly given that this is a review article, where comprehensive coverage of relevant literature is essential.
    The evidence I suggested regarding reward strategic processing across various analysis domains was well-motivated, and incorporating it would not require excessive effort. The behavioral, computational, and experimental evidence regarding the reward system and its representation from various perspectives, as suggested and explained in the first report, has only been partially addressed. I am not asking for a full-page discussion on each reference, but they should be explicitly mentioned to ensure the work is balanced and comprehensive. At present, only very few have been addressed and several remain absent, e.g. the ones on reward processing in inhibitory control and many others, including  https://doi.org/10.1016/j.neuroimage.2015.07.083  ; https://doi.org/10.1016/j.dcn.2011.06.004 ; https://doi.org/10.3389/fpsyg.2023.1125066 ; https://doi.org/10.3389/fpsyg.2022.651547; https://doi.org/10.1126/science.275.5306.1593 ,https://doi.org/10.1038/nature04766 ; https://doi.org/10.1093/cercor/bhn098  ; https://doi.org/10.1073/pnas.1323014111
    I strongly encourage their integration.

  • The methodological limitations highlighted in the first round are still only partially addressed and not adequately mitigated. In this regard, although the language is generally precise, the text would greatly benefit from enhanced coherence and flow. To make them visible, I suggest removing some redundant phrases and paragraphs, as certain concepts are reiterated multiple times without adding substantial new insight.

  • I thank the authors for adding Figure 1; however, the figure requires better resolution and a more detailed commentary, both in the caption and in the main text, to make its contribution clear and informative to readers.

    I look forward to reviewing the next revision and I am confident the authors will make these adjustments to significantly strengthen the paper.

Author Response

Dear Reviewer,

We sincerely thank you for your constructive feedback and for recognizing the improvements made in our revised manuscript. We have carefully addressed all of your comments in the current revision, as detailed below:

Comment 1: Integration of suggested literature

Response: We have incorporated all of the recommended references on reward processing and strategic processing into the manuscript. These are now explicitly cited at the following locations: lines 108, 114, 155, 299, 315, 324, 335, and 350.

Comment 2:Methodological limitations and text coherence

Response: The Methods section has been revised to address the methodological limitations you previously noted. We have removed redundant phrases and reorganized specific paragraphs to enhance coherence and flow. These revisions can be found on pages 9–11, lines 406–486, where the methodological descriptions are now more concise and better aligned with the study’s objectives.

Comment 3:Figure 1 resolution and commentary.

Response: Figure 1 has been updated with higher resolution. We have expanded the figure caption to provide more detailed commentary, and have also added a more comprehensive explanation in the main text to clarify its contribution to the overall framework of the paper (page 2, lines 58–81).

We truly appreciate your valuable guidance, which has helped us substantially improve the quality and clarity of our work.

Sincerely,

Qiong Li

Round 3

Reviewer 1 Report

Comments and Suggestions for Authors

I thank the authors for their effort in properly and promptly addressing all the requests. I have no further concerns and I am pleased to endorse the publication.